# Factors Associated with Surgical Smoke Self-Protection Behavior of Operating Room Nurses

**DOI:** 10.3390/healthcare10050965

**Published:** 2022-05-23

**Authors:** Ching-Lan Yu, Suh-Ing Hsieh, Li-Hung Lin, Shu-Fen Chi, Tzu-Hsin Huang, Shu-Ling Yeh, Chi Wang

**Affiliations:** 1Department of Nursing, Taoyuan Chang Gung Memorial Hospital, Chang Gung University of Science and Technology, Taoyuan City 33378, Taiwan; chinglanyu12002@gmail.com (C.-L.Y.); kathy@cgmh.org.tw (L.-H.L.); nice@cgmh.org.tw (S.-F.C.); q22122@cgmh.org.tw (S.-L.Y.); 2Taiwan Union of Nurses Association, Taipei City 103, Taiwan; grace362966@gmail.com; 3Department of Nursing, Linkou Chang Gung Memorial Hospital, School of Nursing, Chang Gung University, Taoyuan City 33305, Taiwan; gigy@cgmh.org.tw

**Keywords:** surgical smoke, personal characteristics, attitude, perceived attributes, self-protection behavior

## Abstract

Surgical smoke has been proven to be harmful and carcinogenic to humans as well as increasing the risk of acquiring infectious diseases. The operating room nurses’ willingness to use protective equipment against surgical smoke was low. The factors associated with personal protective behavior in the operating room against surgical smoke were sparsely explored. The purpose of this study is to determine factors associated with surgical smoke self-protection behavior of the operating room nurses. This was a descriptive correlational study using a convenience sample from a medical center in northern Taiwan. The self-designed questionnaires included personal characteristics and perceived attributes. The data were analyzed by descriptive and linear regression. Attendance at in-service education with regard to surgical smoke, the attitude to surgical smoke, and surgical smoke self-protection barriers were significant factors found in multivariate linear regression after controlling the covariates. The overall model was significant and accounted for 14.2% of variances. In summary, attending in-service education, attitude and barriers in executing self-protective behaviors were significant factors. It is important to promote operating room nurses’ health through providing correct surgical smoke knowledge, self-protection strategies to improve attitudes toward surgical smoke, improving the hospital’s environment by adding surgical smoke evacuation equipment, and standardizing the operating procedures.

## 1. Introduction

Approximately 10,000 nurses are exposed to surgical smoke each year in Taiwan [1]. The surgical smoke contains blood, chemicals, tissue particles, bacteria, and viruses. which have potential risks of physical, cytotoxic, and genotoxic harm to the personnel in the operating room, [2]. The physical harm includes skin and mucosal diseases (eye, skin, respiratory tract, etc.), central nervous system disorders, infectious diseases (e.g., Human Papillomavirus Virus), and cancer.

The Occupational Safety and Health Act in the US labor law was enacted by the United States in 1970. The Act governs the federal law of occupational health and safety in the private sector and federal government, demonstrating its emphasis on the occupational safety of workers [3]. Moreover, the National Institute for Occupational Safety and Health (NIOSH) and AORN established procedures for controlling surgical smoke during laser/electrosurgical surgery [4] and recommended guidelines for surgical smoke safety [5,6]. Later, the Ministry of Labor of the Republic of China amended the Occupational Safety and Health Act in 2019 [7]. AORN published guidelines regarding surgical smoke safety related to a smoke-free environment, smoke evacuation, education and competency verification, policies and procedures, and quality assurance and performance improvement in 2016 [8]. Furthermore, AORN updated guidelines for surgical smoke safety, including a smoke-free environment, smoke evacuation and filtration, and respiratory protection in 2021 [9]. However, the operating room nurses generally lack knowledge about surgical smoke [10,11], rarely discuss the attitudes toward surgical smoke [12], and mostly fail to take appropriate self-protective behaviors from surgical smoke [11,13,14].

Currently, there are three studies that addressed the relevant factors associated with the self-protective behavior against surgical smoke. They showed that the training on the medical personnel in the operating room [15,16,17,18], size and affiliation of the facility, number of surgical professionals, interactions among medical and nursing personnel, and administrator’s support are related to the self-protective behavior of surgical smoke of medical personnel in the operating room [15,17]. However, there is still a lack of a comprehensive overview of the factors related to the operating room nurses taking self-protective behaviors from surgical smoke. The purpose of this study is to investigate the factors associated with the self-protective behaviors of operating room nurses against surgical smoke. The result could help to raise hospitals’ awareness of the occupational injuries on their staff and provide a safe working environment, encourage the establishment of standard operating procedures, and provide appropriate protective equipment to improve staff health.

## 2. Materials and Methods

### 2.1. Design

This study was a descriptive correlational study with a cross-sectional survey method using a structured questionnaire as a part of a large survey study [19].

### 2.2. Study Framework

The framework of this study is partially based on Rogers’ diffusion of innovations theory, and the independent variables include individual characteristics and perceptions of attributes; the dependent variable is the surgical smoke self-protective behavior (Figure 1). The relationships between individual characteristics, perception attributes, and surgical smoke self-protective behavior were examined.

### 2.3. Participants and Setting

The participants are operating room nurses from two hospitals (a regional teaching hospital and medical center) of the medical care corporate in northern Taiwan using a convenience sampling method. The sample size was estimated using G*Power version 3.1 (Dusseldorf, German) with linear multiple regression, with effect size f2 of 0.15, α of 0.05, power of 0.8, and 41 predictors, with an estimated sample size of 216 participants [20]. Referring to the approximate non-response rate of 30% in most relevant survey studies, the total number of samples required was 281 [12,13,21,22].

### 2.4. Instruments

The instrument for this study was a structured questionnaire on “Operating Room Nurses toward Surgical Smoke Knowledge, Attitude, Self-protective Behavior and Resistance Factors [10,11,12,14,15,16,18,23,24,25,26,27,28]”. Six experts on operating room nursing, toxcicology, occupational medicine, industrial hygiene, environmental science, and environmental engineering and pollution control were entrusted to evaluate the content validity of the questionnaire using a 4-point Likert scale (1 point means highly irrelevant and can be deleted; 4 points means that this topic is highly relevant to the research theme and purpose and does not require any modification). The content of the questionnaire was modified in accordance with the suggestions from these experts. The pilot study was based on 31 operating room nurses of a regional hospital. Three low discrimination questions of the surgical smoke knowledge test were deleted to analyze whether the questions had the function of distinguishing the respondents had correct knowledge of surgical smoke.

#### 2.4.1. Individual Characteristics

Personal characteristics include sex, age, educational level, title, current job department, clinical nursing ladder, existing respiratory problems, etc. (Figure 1).

The original questionnaires on the frequency and severity of symptoms of surgical smoke had 16 items, respectively. According to the opinions of the experts, one item was deleted for epistaxis; the items of nausea and vomiting were combined into one item, and one item was added as an open question for self-reporting other uncomfortable symptoms. The modified version of the questionnaire had 14 items, and the content validity index (CVI) of the questionnaires on the frequency of subjective symptoms and the severity of the questionnaires are 0.99 and 1.00, respectively. The frequency of perceived symptoms was measured on a 4-point Likert scale, with 0 meaning “never” to 3 meaning “always,” and the total possible score range was 0–42. The higher scores indicated the higher frequency of the perceived symptoms. The Cronbach’s α was 0.85 and 0.89 for the pilot study and the present study, respectively. In addition, the severity of self-perception was measured on a 7-point Likert scale, with 0 indicating no symptoms to 6 indicating the most severe symptoms, with a possible total score range of 0–84. The higher scores indicated the more severe of the perceived symptoms. The Cronbach’s α was 0.92 and 0.91 for the pilot study and the present study, respectively.

The original surgical smoke knowledge test was designed as 20 multiple-choice questions with 5 options, referencing the AORN operating room smoke in-service education test [9,29,30,31]. 1 point stands for each correct answer, and 0 point stands for each wrong answer and no idea. According to the opinion of the experts, four questions with unclear meaning and not related to professional knowledge were deleted, and three questions with low discrimination were removed, leaving 13 questions with a possible total score range of 0–13. The higher scores indicated the higher knowledge score. The CVI before and after the revision were both 0.74, and the Kuder-Richardson-20 (KR-20) after the deletion was 0.44, with the test difficulty level of 0.54.

#### 2.4.2. Perceptions of Attributes

The original surgical smoke attitude questionnaire consisted of 11 items, including nurses’ attitudes toward the protective equipment and in-service education provided by the hospital, their attitudes toward the use of protective equipment, and the hazards of surgical smoke, on a 4-point Likert scale from 1 meaning “strongly disagree” to 4 meaning “strongly agree,” with a total score range of 11–44. The higher score indicates the higher attitude score toward surgical smoke. After revising the questionnaire according to the suggestions from the experts, the CVI was 0.81, and the Cronbach’s α of the pilot study and the present study were 0.72 and 0.88, respectively.

The original surgical smoke self-protection resistance factor questionnaire consisted of seven items, mainly for understanding the reasons for not using protective equipment and materials, on a 4-point Likert scale from 0 for “never” to 3 for “always”. According to the suggestions from the experts, the wall-mounted and mobile suction systems were compared separately and one question was added for the open-ended question “other resistance factors”. The revision of the questionnaire consisted of 9 items with a possible total score range of 0–27. The higher score represents higher resistance scores for self-protection against surgical smoke. The CVI was 0.76, and the Cronbach’s α of the pilot study and the present study were 0.70 and 0.71, respectively.

#### 2.4.3. Self-Protective Behaviors for Exposure to Surgical Smoke

The original surgical smoke self-protective behavior questionnaire consisted of six questions, including the frequency of using the most frequently used protective equipment and objects, on a 4-point Likert scale from 0 for “never” to 3 for “always”, with a possible total score range of 0–18. The higher score represents higher scores of surgical smoke self-protective behavior. There is another question on the reasons for using wall-mounted or mobile suction systems. According to the suggestions from the experts, the wording of the questions in this questionnaire was revised and one question was added. The modified version of the questionnaire was seven items with a CVI of 0.81, and the Cronbach’s α was 0.65 and 0.39 for the pilot study and the present study, respectively.

### 2.5. Data Collection

The pilot study was conducted from 2 October to 9 October 2019, recruiting 31 nurses in a regional teaching hospital. The researcher explained the purpose, methodology, and rights of the participants to the operating room nurses. The questionnaires were distributed to those who agreed to participate. The questionnaire takes approximately 20 min to complete and a $250 gift voucher would be given upon completion of the questionnaire. A total of 290 questionnaires were distributed in this study, and 290 questionnaires were collected, with a recovery rate of 100%, of which 283 (97.59%) were valid.

### 2.6. Ethical Considerations

This study was approved by the Institutional Review Board (Case No. 201900857B0). Before beginning our research to distribute the study instructions and questionnaires, informed consent was obtained from the operating room nurses. They were informed that they can withdraw from the research at any time.

### 2.7. Data Analysis

The analysis was conducted using SPSS 21.0 (IBM, Armonk, NY, USA). The continuous variables were checked for assumptions such as normality, extreme values, and multicollinearity, and extreme values were removed by using winsorizing. The statistical methods included descriptive and bivariate and multivariate linear regressions. The significant level was defined as *p* < 0.05. The variables with *p* < 0.25 in the bivariate analysis were entered into the multivariate linear regression analysis of individual characteristics and perceptions of attributes, and the variables of two blocks with *p* < 0.25 were entered into the overall multivariate linear regression analysis [32].

## 3. Results

### 3.1. Individual Characteristics

95.1% of the operating room nurses were female, with an average age of 35.81 years (SD = 8.42), and the highest proportion of the academic program was a two-year technical program (40.3%) (Appendix A: Figure A1); 91.5% did not learn the concept of “surgical smoke” during their school years and 68.2% attended in-service education on the concept of “surgical smoke” during employment, with an average of 2.00 years (SD = 8.37); the highest proportion of the clinical nursing ladder was N4 (34.3%) (Appendix A: Figure A2), most of the operating room nurses were unidisciplinary, daily exposure to surgical electrocautery smoke averaged 6.73 h (SD = 2.26) in the operating room, and wall-mounted or mobile suction systems were the most frequently used in general surgery (42.8% vs. 39.9%). The total frequency of subjective symptoms from nurses in the operating room exposed to surgical smoke ranged from 0–25 with a mean of 9.02 (SD = 6.00), and the total severity of subjective symptoms ranged from 0–53 with a mean of 14.95 (SD = 12.06). The total score for the knowledge of surgical smoke ranged from 0–13 with a mean of 6.93 (SD = 2.16) and a correct response rate of 53.3% (Table 1).

### 3.2. Perceptions of Attributes

The total attitude score of the operating room nurses towards surgical smoke ranged from 27–44 with a mean of 39.41 (SD = 4.00) (Table 1). The rate of operating room nurses who checked “agree” and “strongly agree” in the 11 items on the attitude toward surgical smoke was significantly high, but among which, “to wear an N95 mask” had the lowest rate in terms of “agree” (41.3%) and “strongly agree” (29.0%).

The total score of resistance factors for self-protection against surgical smoke among operating room nurses ranged from 0–22, with a mean of 9.23 (SD = 3.89) (Table 1). Among the nine items on resistance to surgical smoke self-protection by operating room nurses, the top three in terms of frequency (the total rate of sometimes, often, and always) were that wearing an N95 mask would make breathing difficult (96.1%), the mobile suction system too noisy (91.8%), and the wall-mounted suction system too noisy (86.6%).

### 3.3. Self-Protective Behaviors When Exposure to Surgical Smoke

The total score of self-protective behaviors for surgical smoke among operating room nurses ranged from 4 to 20, with a mean of 10.29 (SD = 2.95) (Table 1). The top 2 highest rates of frequently and always using surgical smoke self-protective behaviors among operating room nurses were surgical masks (77.7% vs. 13.4%) and wall-mounted suction systems (41.0% vs. 26.1%), and the bottom two lowest rates were N95 masks (1.4% vs. 4.6%), and laser masks (4.6% vs. 9.2%).

### 3.4. Factors Associated with Surgical Smoke Self-Protective Behaviors among Operating Room Nurses

The bivariate regression analysis in Appendix B (Table A1) showed that: current working department (≥two departments vs. single department), use of a wall-mounted suction system for fracture reconstruction, use of mobile suction system for other procedures, and surgical smoke attitude score were significant correlated. However, after controlling for covariates, the multivariate regression analysis showed no in-service education on surgical smoke concepts during employment (b = −0.77, *p* = 0.038), attitude toward surgical smoke (b = 0.15, *p* = 0.001), and resistance to surgical smoke self-protective behaviors (b = −0.10, *p* = 0.028) were significantly correlated factors. The overall model was significant (*F*_(18,263)_ = 3.59, *p* < 0.001), explaining 14.2% of the variance in the nurses’ surgical smoke self-protective behavior (Adjusted *R*^2^ = 0.142).

## 4. Discussion

### 4.1. Individual Characteristics

This study found that drowsiness, headache, runny nose or other nose discomforts and cough, dizziness, and tearing or other eye discomforts were the top six common symptoms in the operating room (Appendix A: Figure A3), which are similar to the findings of Asdornwised et al. [28], Ball & Gilder [33], Ilce et al. [14], and Addley [22], but with different orders. In comparison with other studies, the most common symptom experienced with surgical smoke was headache. The second was respiratory symptoms and eye symptoms. Asdornwised et al. [28] showed that coughing and sneezing were the most serious, and headache was the most severe symptom of this study. This was caused by the high concentrations of surgical smoke that irritated the upper respiratory tract for the operating room nurses, and the same as with smokers with regard to the potential increase of the incidence of headache [34]. This may result from the fact that surgical smoke increases the sensitivity of the brain’s pain receptors, narrows blood vessels, and reduces blood flow to the brain, decreasing the effectiveness of pain medications and making pain relief more difficult [35].

This study found that the total score of surgical smoke knowledge ranged from 0–13 with a mean of 6.93 ± 2.16. This was higher than that of Arli [36], which showed that the total score of surgical smoke knowledge ranged from 2–10 with the mean of 5.19 ± 1.46 (5.10 ± 1.56 for operating nurses). However, unlike Arli’s study, the number of surgical smoke knowledge questions was 16, and the participants included surgeons, anesthesiologists, surgical technicians, anesthetists, and operating room nurses (66.1%). The number of surgical smoke questions in this study was 13 and there were only 283 operating room nurses recruited in the study. The knowledge of surgical smoke among the participants were both low in Arli’s and our series. Therefore, hospital administrators should improve the knowledge of surgical smoke protection among operating room nurses through in-service education and training.

### 4.2. Perceptions of Attributes

The participants agreed or strongly agreed with the attitude towards surgical smoke, with a rate of 96% or more. 99.3% of the participants thought that appropriate protective measures should be taken to avoid the harm of surgical smoke. But when it came to wearing N95 masks during surgery, only 70.3% of the participants agreed or strongly agreed, indicating that the participants would choose to take self-protective measures under certain conditions. It is possible that the damage of surgical smoke gradually affects the health of operating room nurses, unlike COVID-19, which has the risk of being transmitted. Owing to no immediate threat of surgical smoke, the operating room nurses feel difficulty in breathing due to wearing N95 masks, thereby causing the lower score of wearing N95 masks. Therefore, the hospital administrators can enhance the knowledge and attitude of surgical smoke protection through in-service education and training, or develop low-cost, well-ventilated, and effective masks to improve the health of operating room nurses.

In the studies of Ball [16], Edwards & Reiman [24], and Asdornwised et al. [28], they noted that physicians’ refusal to using smoke evacuators and the inconvenience due to the loud noise of the evacuators serve as barriers to the use of smoke evacuators. The results of this study showed that noisy wall-mounted evacuators and mobile suction systems were the highest causes affecting surgical smoke self-protective behavior, while physicians’ refusal to use smoke evacuators was the least common factor. Noise pollution (>75 decibels, dB) is second only to air pollution (surgical smoke) and can have various effects on the health of operating room staff, including physical effects such as rapid breathing and heart rate, high blood pressure, annoyance, headache, memory disorder and poor concentration, and cardiovascular and metabolic systems disorders and psychological aspects such as attacks of stress, fatigue, depression, anxiety, and cognitive impairment [37,38]. The noise of the smoke evacuators makes the surgeons unable to concentrate on the surgical procedure, which affects their judgment during surgery and endangers the safety of the patient [38]. Therefore, the operating room should use low-noise smoke evacuators and regular maintenance of the smoke evacuators.

### 4.3. Self-Protective Behavior from Exposure to Surgical Smoke

This study found that the top two self-protective behaviors for surgical smoke among operating room nurses were surgical masks (91.1%) and wall-mounted suction systems (67.1%), while the lowest two were N95 masks (6.0%) and laser masks (13.8%). The lowest rate in wearing N95 masks for operating room nurses might be because of the difficulty in breathing after wearing it. This corresponded with the result of the attitude toward surgical smoke in the survey question “medical and nursing personnel should wear N95 masks during surgery to prevent surgical smoke hazards”, which has the highest rate in choosing “strongly disagree” and “disagree”. This study found that the rate of wearing N95 masks and goggles was higher than the study by Asdornwised et al. [28]. In their study performed in Thailand, the rate of “often” and “always” in properly wearing high filtration surgical masks was 25.2% and 16.7%, respectively, which were lower than for those concerning protective eyewear (29.2% and 25.2%, respectively). However, in the current study, 68.2% of the participants attended in-service education on “surgical smoke” during their employment, which was also higher than that in the study by Asdornwised et al. [28]. The potential reason was that the participants of the Asdornwised et al. [28] study were older, and they had higher education levels and had more years of experience in the operating room than the participants of the current study; moreover, they would acquire knowledge about surgical smoke through colleagues, mass media and other sources. In addition, this study found that wearing surgical masks (97.5% vs. 91.1%), N95 masks (66.8% vs. 8.9%), and goggles (69.3% vs. 46.7%) was higher than that found by Ilce et al. [14] in a hospital in Turkey, which may be caused by the fact that all the participants in this study had more years of experience in the operating room than the participants in the study by Ilce et al. [14].

### 4.4. Factors Affecting Surgical Smoke among the Operating Room Nurses 

The purpose of this study was to examine the factors associated with the self-protective behaviors of operating room nurses against surgical smoke. In this study, “no in-service education on surgical smoke concepts during employment”, “surgical smoke attitude score”, and “resistance factor score for surgical smoke self-protection” were found to be significantly correlated with self-protective behaviors. In other words, those who did not participate in in-service education on surgical smoke concepts had better self-protective behaviors than those who did. The surgical smoke attitude has the strongest influence on the self-protective behavior of operating room nurses with a standardized b of 0.15; that is, every 1 point increase in surgical smoke attitude would increase the self-protective behavior by 0.15 points; every 1 point increase in resistance would decrease the self-protective behavior score by 0.10 points.

Firstly, “those who did not attend in-service education on surgical smoke concepts during employment had higher self-protective behaviors than those who did”, is different from the findings of Ball [16] and Steege et al. [18]. This study showed that those who did not attend in-service education on surgical smoke concepts had higher self-protective behaviors than those who did. This could be the reason that 56.8% of the participants have worked in the operating room for over 10 years. It is easy to develop protective behaviors based on their work experience. The top five “always” adopted self-protective behaviors were surgical masks (77.7%), wall-mounted suction systems (41.0%), general medical masks (36.4%), mobile suction systems (26.1%), and goggles (8.1%). The surgical masks generally filter particles size of 5 μm or larger, while the surgical smoke particles from electrocautery are 0.07–0.42 μm. Thus, surgical masks are not suitable for filtering electrocautery smoke particles, indicating that the overall self-protective behavior of nurses is inadequate [14,21]. The studies by Massarweh et al. [39] and Spearman et al. [13] showed the same low rate for the personal use of protective equipment. The rate of selecting the correct answers on the knowledge test was 53.3%, indicating that the participants’ knowledge of surgical smoke is inadequate. Therefore, it is important to consider how to alter their attitudes and implement self-protective behaviors in clinical practice through an innovative interventional in-service education program.

Secondly, “surgical smoke attitude” had the strongest influence on the self-protective behaviors of operating room nurses. In other words, higher surgical smoke attitude scores indicated better-implemented behaviors. This aligned with the studies from Ball [15,17] that operating room nurses with positive attitudes towards surgical smoke evacuation recommendations are more likely to implement surgical smoke evacuation recommendations. The rate of “agree” and “strongly agree” among the operating room nurses ranged from 96.5% to 100%. Except for “medical personnel should wear N95 masks during surgery, “the rate of “agree” and “strongly agree” was 70.3%, showing that nurses conditionally chose self-protective behavior. In addition to enhancing education and strengthening attitudes, the management team should develop new products or find alternatives with the same protection because respiratory protection is the last line of defense to prevent nurses from surgical smoke exposure.

Lastly, the rate of “resistance factor score of surgical smoke self-protective behavior” has the opposite trend of “self-protective behavior score”. In this study, the most common occurrence of resistance factor was “wearing N95 mask makes breathing difficult”, which is in line with the 70.3% who “agree” and “strongly agree” with the surgical smoke attitude that “medical personnel should wear N95 masks during surgery”. To avoid the harm of surgical smoke, in addition to regular reviews of the efficiency and maintenance of the high-efficiency particulate air filter (HEPA) system and smoke evacuator in the operating room by the management, a policy on surgical smoke evacuation should also be established. For example, Rhode Island was the first state in the United States to pass legislation requiring the use of smoke evacuators in every surgery that produces surgical smoke [40]. It is believed that with government regulation, support and oversight from hospital management and implementation by operating room nurses, surgical smoke will be eliminated to achieve a smoke-free working environment.

This study has several limitations. First, this study used a non-probability sampling method and recruited operating room nurses from a medical foundation. Therefore, the findings of this study were not generalizable to other hospitals. Second, the internal consistency of the self-protective behavior questionnaire between the pilot study and the present study were 0.65 and 0.39. This might be caused by the different characters of the two hospitals. Third, the overall model of multivariate linear regression can only account for 14.2% of the variance in the nurses’ surgical smoke self-protective behavior that might need to explore other factors associated with self-protective behaviors of operating room nurses for the future studies. Fourth, this study was conducted at two hospitals, so that the organization characteristics (e.g., type, size, leadership support) and the expectation of the staff on surgical smoke protection were not explored.

## 5. Conclusions

“No in-service education on surgical smoke concepts during employment”, “surgical smoke attitude score”, and “resistance factor score for surgical smoke self-protection”, were significantly correlated with self-protective behaviors. By increasing nurses’ self-awareness of surgical smoke hazards, these factors can influence the surgical team in terms of putting greater emphasis on using self-protective gear properly and implementing surgical smoke evacuation policies. Moreover, through the innovative design of easy-to-breathe, protective, and low-cost surgical smoke protection devices, they can enhance the self-protective behavior of medical personnel in the operating room, establish a smoke-free workplace, and reduce smoke exposure to improve surgical quality and maintain the health of the surgical teams and patients. The hospital directors of the nursing department and the hospital administration need to watch out for the safety and health of employees in the operating room by establishing smoke evacuation policies, monitoring surgical smoke self-protective behaviors of employees in the operating room, and providing adequate and sufficient surgical smoke equipment and utilities.

## Figures and Tables

**Figure 1 healthcare-10-00965-f001:**
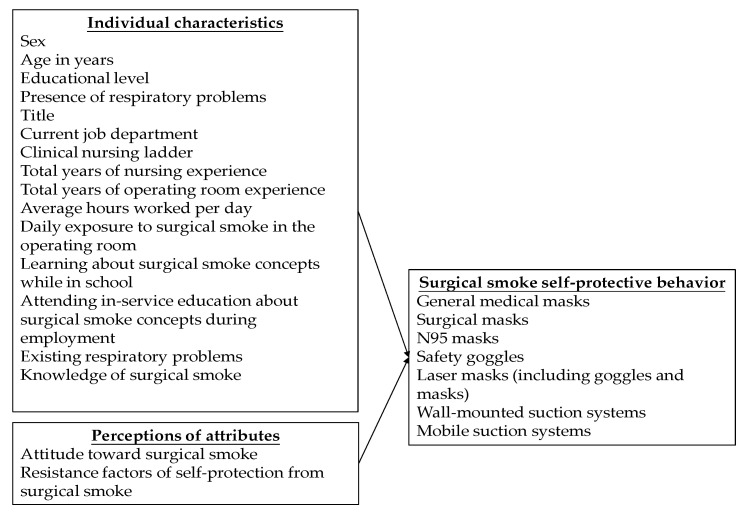
The study framework.

**Table 1 healthcare-10-00965-t001:** Score of the surgical smoke knowledge test, perceptions of attributes, and self-protective behaviors for surgical smoke in the operating room nurses.

Instruments	# Items	Rating Scale	Range	Mean (SD)
The frequency of symptoms of surgical smoke	14	0–3	0–25	9.02 (6.00)
The severity of symptoms of surgical smoke	14	0–6	0–53	14.95 (12.06)
The surgical smoke knowledge test	13	0–1	0–13	6.93 (2.16)
The surgical smoke attitude	11	1–4	27–44	39.41 (4.00)
The surgical smoke self-protection resistance factor	9	0–3	0–22	9.23 (3.89)
The self-protective behaviors for surgical smoke	7	0–3	4–20	10.29 (2.95)

## Data Availability

The datasets used and analyzed during the current study are available from the first author and the corresponding authors on reasonable request.

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
