# Peer review of "Factors Associated with Surgical Smoke Self-Protection Behavior of Operating Room Nurses"

_healthcare, 2022, doi:10.3390/healthcare10050965_

Round 1
Reviewer 1 Report
The paper presented an intresting problem for the health of the nurses in operating room. The introduction , material and methods and conclusion are well presented. Results is not very well presented . The results can be presented as a graphics. The ilustration of the results will improve the work and will make it more readable and understanding.
Author Response
Point 1: The paper presented an interesting problem for the health of the nurses in operating room. The introduction, material and methods and conclusion are well presented.
Response 1: Thank you for your feedback.
Point 2: Results is not very well presented. The results can be presented as a graphics. The illustration of the results will improve the work and will make it more readable and understanding.
Response 2: The introduction (pp.1-2 line 42-44,47, 53-59, 63-64, 66-68,70, &72-74), methods (p.2 line 80 & 86-87; p.3 line 103-105; 111, 119-122; p.4 line 144; 153, 157-158, 170, 176 ; p.5 line 191, 193-194 &196), results (p.5 line 213, 217-218, 222-223, 230 ; p.6 line 234-235, & 240), discussion (p.6, line 254-256,269, & 275-276 ; p.8 line 314-316,320-321,323 &328; p.9 line 398-401), conclusions (p.9 line 413-417), and references (p.10 line 454, 459-460; p.11 line 498-499) have reviewed and revised. Table 1 was added to the main text instead of graphics, because graphics cannot present all individual characteristics and perceptions of attributes due to the restriction of number of wording characters.

Reviewer 2 Report
The work entitled: Factors Associated with Surgical Smoke Self-Protection Behavior of Operating Room Nurses, is well written, the description of the methodology is correct and perhaps a bit extensive, the subject is of interest to improve hygiene conditions in the surgical environment.
Other works, which the authors cite, address these problems in a different context, affecting their originality.
It would be preferable to generate graphic material that allows the reader to understand the data described more pleasantly.
Author Response
Point 1: The work entitled: Factors Associated with Surgical Smoke Self-Protection Behavior of Operating Room Nurses, is well written, the description of the methodology is correct and perhaps a bit extensive, the subject is of interest to improve hygiene conditions in the surgical environment..
Response 1: Thank you for the comment.
Point 2: Other works, which the authors cite, address these problems in a different context, affecting their originality.
Response 2: Appreciate all your effort for giving comments.
Point 3: It would be preferable to generate graphic material that allows the reader to understand the data described more pleasantly.
Response 3: The introduction (pp.1-2 line 42-44,47, 53-59, 63-64, 66-68,70, &72-74), methods (p.2 line 80 & 86-87; p.3 line 103-105; 111, 119-122; p.4 line 144; 153, 157-158, 170, 176 ; p.5 line 191, 193-194 &196), results (p.5 line 213, 217-218, 222-223, 230 ; p.6 line 234-235, & 240), discussion (p.6, line 254-256,269, & 275-276 ; p.8 line 314-316,320-321,323 &328; p.9 line 398-401), conclusions (p.9 line 413-417), and references (p.10 line 454, 459-460; p.11 line 498-499)have reviewed and revised. Table 1 was added to the main text instead of graphics, because graphics cannot present all individual characteristics and perceptions of attributes due to the restriction of number of wording characters.

Reviewer 3 Report
Dear authors, I consider the topic of your research very important because by arousing nurses' self-awareness of surgical smoke hazards, they can influence the surgical team to put more emphasis on using self-protective gears properly and the implementation of surgical smoke evacuation policies. However, I think it's worth presenting the prevention methods in more detail. It is also worth describing what are the expectations of the staff and the tasks for hospital directors.
Author Response
Point 1: Dear authors, I consider the topic of your research very important because by arousing nurses' self-awareness of surgical smoke hazards, they can influence the surgical team to put more emphasis on using self-protective gears properly and the implementation of surgical smoke evacuation policies.
Response 1: Thank you for the comments.
Point 2: However, I think it's worth presenting the prevention methods in more detail.
Response 2: The prevention methods of surgical smoke have added information to the introduction with guidelines (pp.2 line 54-59).
Point 3: It is also worth describing what are the expectations of the staff and the tasks for hospital directors.
Response 3: The expectation of the staff and the tasks for hospital directors have added to the fourth limitation (p.9 line 398-401). The conclusions have addressed the tasks of hospital directors (p.9 line 413-417).
